# Quantitative Proteomics Reveals Significant Differences between Mouse Brain Formations in Expression of Proteins Involved in Neuronal Plasticity during Aging

**DOI:** 10.3390/cells10082021

**Published:** 2021-08-07

**Authors:** Dominika Drulis-Fajdasz, Kinga Gostomska-Pampuch, Przemysław Duda, Jacek Roman Wiśniewski, Dariusz Rakus

**Affiliations:** 1Department of Molecular Physiology and Neurobiology, University of Wrocław, Sienkiewicza 21, 50-335 Wrocław, Poland; dominika.drulis-fajdasz@uwr.edu.pl (D.D.-F.); przemyslaw.duda@uwr.edu.pl (P.D.); 2Biochemical Proteomics Group, Department of Proteomics and Signal Transduction, Max Planck Institute of Biochemistry, 82152 Martinsried, Germany; kinga.gostomska-pampuch@hirszfeld.pl (K.G.-P.); jwisniew@biochem.mpg.de (J.R.W.); 3Department of Biochemistry and Immunochemistry, Wrocław Medical University, Chałubińskiego 10, 50-368 Wrocław, Poland

**Keywords:** glutamatergic and GABAergic transmission, Camk2, OXPHOS, extracellular matrix, total protein approach, hippocampus, cortex, cerebellum

## Abstract

Aging is associated with a general decline in cognitive functions, which appears to be due to alterations in the amounts of proteins involved in the regulation of synaptic plasticity. Here, we present a quantitative analysis of proteins involved in neurotransmission in three brain regions, namely, the hippocampus, the cerebral cortex and the cerebellum, in mice aged 1 and 22 months, using the total protein approach technique. We demonstrate that although the titer of some proteins involved in neurotransmission and synaptic plasticity is affected by aging in a similar manner in all the studied brain formations, in fact, each of the formations represents its own mode of aging. Generally, the hippocampal and cortical proteomes are much more unstable during the lifetime than the cerebellar proteome. The data presented here provide a general picture of the effect of physiological aging on synaptic plasticity and might suggest potential drug targets for anti-aging therapies.

## 1. Introduction

Physiological aging is related to a gradual decline in cognitive functions, such as memory formation and retention, processing speed and conceptual reasoning [1]. It is widely accepted that aging-associated changes are caused mainly by the loss of neuronal cells; however, numerous studies have suggested that the neuronal network structure, rather than the number of neurons, is affected during aging [2,3,4]. It has been demonstrated that aging is associated with the shortening of dendrites and a decrease in their number, loss of dendritic spines, a decrease in the axon number within a network and the level of their myelination and loss of synapses [5]. Additionally, it has been shown that the molecular mechanisms underlying brain plasticity phenomena such as long-term potentiation and depression of the glutamatergic (LTP and LTD) and GABAergic (iLTP and iLTD) synapses and the efficacy of various neuromodulator systems decline with advancing age [6,7]. 

Several studies have demonstrated that aging-related cognitive changes are accompanied by changes in the expression and/or localization of proteins involved in synaptic transmission and plasticity [6,8]. The expression of proteins in various regions of the rodent brain has been intensively studied by mass spectrometry-based proteomic techniques [9,10] and microarrays [11,12]. However, except for Walter and Mann’s study [10] and Duda et al.’s study [8], all the investigations employed semiquantitative approaches which do not deliver information about the accurate concentration of proteins in the studied samples.

Recently, we measured concentrations of proteins involved in neuronal plasticity in the hippocampus, cerebral cortex and cerebellum in young (1-month-old) and adult (12-month-old) mice [8]. We found that, while the total amount of proteins did not change during the lifetime, the neurotransmission- and neuroplasticity-related protein titers differed significantly between young and adult animals, which indicates that the symptoms of signal transmission and neuroplasticity weakening may be observed in middle-aged mice at a proteomic level [8].

In this paper, we provide the results of an in-depth quantitative analysis of neuronal plasticity-related hippocampal, cortical and cerebellar proteomes of young (1-month-old) and old (22-month-old) mice, focusing mainly on aging-related changes in the concentration of proteins engaged in glutamate, GABA, acetylcholine and monoamine signaling. We also discuss the changes in the concentration of proteins involved in neurotransmitter release and formation of synaptic connections, such as proteins of trans-synaptic cell adhesion and perineuronal nets. 

Using the label-free total protein approach (TPA) method, we measured the titers of more than 7000 proteins in each of the studied brain regions [13]. In this paper, we present, thus far, the most in-depth quantitative proteomic description of synaptic plasticity-related changes during the physiological aging of mice. To the best of our knowledge, this is, thus far, the most in-depth quantitative proteomic description of synaptic plasticity-related changes during the physiological aging of mice.

## 2. Materials and Methods

### 2.1. Animals and Preparation of Tissues

Brains were isolated from five female C57BL/10J mice at P30 (young) and five 22-month-old (aged) mice. Animals were treated as described in [8]. Briefly, the animals were anesthetized with isoflurane and decapitated, and brains were explanted in an ice-cold buffer (87 mM NaCl, 2.5 mM KCl, 1.25 mM NaH_2_PO_4_, 25 mM NaHCO_3_, 0.5 mM CaCl_2_, 7 mM MgSO_4_, 25 mM glucose, 75 mM sucrose, pH 7.4). The whole brain regions: right hippocampus, frontal cortex from the right hemisphere and right hemisphere of the cerebellum, from each animal were used for quantitative proteomics. All the procedures were approved by the local ethics committee (Wroclaw Ethical Committee, permission no.10/2018), and every effort was made to minimize the number of animals used for the experiments.

### 2.2. Preparation of Tissue Lysates

The lysates were obtained as described in [13]. The isolated structures were homogenized in lysis buffer (0.1 M Tris/HCl, 2% SDS, 50 mM DTT, pH 8.0) and incubated for 5 min at 99 °C. Samples were stored at −20 °C until proteomic analysis. Tryptophan fluorescence was employed to determine total protein concentration in the samples [14].

### 2.3. Multi-Enzyme Digestion Filter-Aided Sample Preparation (MED FASP)

The lysates containing 80 µg of total protein were used for the MED FASP [15] without alkylation of cysteine [16]. The proteins were cleaved overnight with LysC and then digested with trypsin for 3 h. The enzyme-to-protein ratio was 1:40. Digestions were carried out at 37 °C in 50 mM Tris-HCl with the addition of 1 mM DTT, pH 8.5 [13]. Aliquots containing 8 µg of total peptide were concentrated and stored at −20 °C until mass spectrometry analysis.

### 2.4. Liquid Chromatography-Tandem Mass Spectrometry

The analysis of peptide mixtures was performed as described earlier [13] using the QExactive HF mass spectrometer (ThermoFisher Scientific, Palo Alto, CA, USA). The data were deposited in the ProteomeXchange Consortium via the PRIDE partner repository [17]. The dataset identifier: PXD025978 (username: reviewer_pxd025978@ebi.ac.uk; password: QL3YI7nP).

### 2.5. Proteomic Data Analysis

MaxQuant v1.2.6.20 [18] was used for the MS data analysis. The UniProtKB/Swiss-Prot database was employed to identify the proteins using MS and MS/MS peptide data. 

Carbamidomethylation of cysteine was set as a fixed modification. The initial allowed mass deviation of the precursor ion was up to 6 ppm, and for the fragment masses, it was up to 20 ppm. The maximum false peptide discovery rate was specified as 0.01. The total protein approach method [19,20] was used to calculate the protein molar concentration using the relationship
$$c\left(i\right)=\frac{MS_{signal}\left(i\right)}{total\;MS_{signal}\times MW\left(i\right)}\;\left[\frac{mol}{g\;total\;protein}\right]$$
where the amount of individual proteins (***c***(***i***)) was calculated as the ratio of their intensity (***MS_signal_***(***i***)) to the sum of all intensities (***total MS_signal_***) in the measured sample, multiplied by the molecular weight (***MW***(***i***)) of individual proteins.

### 2.6. Statistical Analysis

Data are presented as mean ± SD. The equality of variances was calculated using the Fisher F-test. To determine the differences between any two experimental groups, Student’s *t*-test was used. The analysis was performed using SigmaPlot 11 software (Systat Software).

## 3. Results

The full proteomic data were deposited in the ProteomeXchange Consortium, and they are available with the dataset identifier: PXD025978 (username: reviewer_pxd025978@ebi.ac.uk; password: QL3YI7nP). The analysis of the spectra allowed for quantitation of 7547 proteins across the analyzed samples. Proteins which were identified with at least one unique peptide were used in the analysis. In detail, the expression data for 7324, 7088 and 7343 proteins in, respectively, the hippocampus, cortex and cerebellum of old animals were quantitatively determined and deposited to the ProteomeXchange Consortium via the PRIDE partner repository. For young animals, the data for 7347, 7323 and 7526 proteins in, respectively, the hippocampus, cortex and cerebellum were deposited therein.

## 4. Glutamatergic Transmission

Glutamate, the main excitatory neurotransmitter in the brain, acts through ionotropic and metabotropic receptors (mGluRs, Grm). The ionotropic receptors fall into one of four classes: α-amino-3-hydroxy-5-methyl-4-isoxazolpropionic acid receptors (AMPA receptors, Gria), *N*-methyl-*D*-aspartate receptors (NMDA receptors, Grin), kainate receptors (Grik) and delta receptors (Grid).

### 4.1. Gria

Our study reveals that the most abundant glutamate receptors in the hippocampus are Gria. They are heterotetrameric proteins [21], and we found that the Gria2 subunit was expressed at the highest titer both in young and in aged murine hippocampi (Figure 1A). Aging had no effect on the expression of Gria receptors in the hippocampus, except for Gria4, whose level was significantly reduced in old animals. However, the titer of Gria4 was very low in general, as compared to other members of the Gria family.

In the cerebral cortex, Gria were the most ubiquitous glutamate ionotropic receptors only in the old animals (Figure 1B,D). Statistically, the changes in the titers of individual Gria subunits during aging in the cortex were not significant, but the total Gria protein concentration was significantly increased in old animals (Figure 1B). Similar to the hippocampus, Gria2 was the main isoform in the cortex. 

In the cerebellum, the most abundant glutamate receptors were Gria1 and Gria2, which were expressed at very similar levels, and aging did not affect their concentration (Figure 1C).

### 4.2. Grin

NMDA receptors are glutamate-dependent heterotetrameric channels for calcium and sodium ions which are crucial for synaptic plasticity [22] and are composed of Grin isoforms [23]. We found that the main form of Grin expressed in all the studied brain structures was Grin1 (Figure 1A–C). The concentration of this isoform was not affected by aging in the hippocampus and cerebellum; however, it was significantly reduced in the cortex (Figure 1B). Overall, the total concentration of all NMDA subunits, except for Grin3a (whose level was unaffected by aging and expressed at a very low level as compared to other Grin proteins), was decreased in the aged cortex, while it was unchanged in the cerebellum (Figure 1B,C). In the hippocampus, aging was reflected by a statistically significant decrease in the total amount of Grin proteins, and also the Grin2b and Grin3a isoforms.

### 4.3. Grid

Grid1 and Grid2 are ionotropic receptors expressed almost exclusively in the cerebellum, and they are involved in the regulation of synaptic plasticity [24]. We found that in the cerebellum, the main isoform is Grid2, and that its titer is not affected by aging (Figure 1C). In the hippocampus and cortex, the concentration of, respectively, Grid1 and Grid2 is significantly lower in aged animals; however, the titer of these receptors is very low in general (Figure 1A,B). 

### 4.4. Grik

Kainate receptors are heteromeric sodium channels mediating ionotropic and metabotropic transmission [25,26]. Our analysis revealed that the concentration of almost all members of the Grik protein family was decreased in the old hippocampi and cortices (Figure 1A,B). In contrast, the titer of Grik proteins, except for the Grik2 isoform, was unaffected by aging in the cerebellum (Figure 1C).

### 4.5. Grm

We identified seven members of glutamate-dependent metabotropic receptors, Grm, in all the studied brain formations: Grm1–7 (Figure 1A–C). The total concentration of Grm proteins in the hippocampus and cortex was reduced in aged animals (Figure 1A,B), while the total titer of cerebellar Grms was not significantly changed by aging (Figure 1C).

Detailed analysis revealed that the levels of the most abundant isoforms of Grm (Grm2, Grm3 and Grm5) in the hippocampus were significantly decreased in old mice (Figure 1A), while in the cortex, the changes in the titers of the most abundant isoforms were not significantly altered (Figure 1B).

## 5. GABAergic Transmission

γ-Aminobutyric acid is the most abundant inhibitory neurotransmitter in the brain, and it can interact with two types of receptors: the ionotropic GABA_A_ receptor, which is a chloride channel, and the metabotropic GABA_B_ receptor. GABA_A_ receptors are pentameric proteins composed of various subunits: α (Gabra), β (Gabrb), γ (Gabrg) and δ (Gabrd) [27]. GABA_B_ is a heterodimeric G protein-coupled receptor formed by the Gabbr1 and Gabbr2 subunits [28]. 

## 6. GABA_A_–Gabr

Our analysis demonstrated that the total concentration of the GABA_A_ protein was significantly reduced in the hippocampus and cortex of old mice, but it was not affected by aging in the cerebellum (Figure 2A). These changes were reflected by alterations in the expression of individual subunits of the GABA_A_ receptor (Figure 2B–D).

### 6.1. Gabra (subunit α)

We found that that the overall concentration of Gabra subunits was not affected by aging in all the studied brain formations (Figure 2B–D). The most abundant isoform was Gabra1. Within the α subunits, only Gabra3 and Gabra5 expression was influenced by aging (Figure 2B–D). The only statistically significant change was associated with Gabra4, whose titer was decreased in the cortex of aged animals (Figure 2C).

### 6.2. Gabrb (subunit β)

Our analysis showed that subunit β was the most ubiquitous GABA_A_ subunit in all studied brain structures, both in young and in old animals (Figure 2B–D), and the main isoform of the β subunit was Gabrb2 (Figure 2B–D). The concentration of Gabr2b was the highest in the cerebellum, while in the hippocampus and cortex, its titers were similar (Figure 2B–D). We did not observe any significant aging-associated changes in the expression of isoforms of the β subunit in the hippocampus and cerebellum. However, in the cortex, the total titer of Gabrb was significantly reduced, and this was related to a decrease in the Gabrb2 titer (Figure 2C).

### 6.3. Gabrg (subunit γ)

Among the γ subunits of the GABA_A_ receptor, Gabrg2 was expressed at the highest level in the cortex and cerebellum (Figure 2C,D), and it was the only γ subunit we could detect in the hippocampus (Figure 2B). In the hippocampus and cortex, the total Gabrg isoform concentration was almost two times higher in young animals than in aged ones (Figure 2A,B). On the other hand, the total γ subunit concentration was not affected by aging in the cerebellum (Figure 2D).

### 6.4. Gabrd (subunit δ)

In old mice, the δ subunit was expressed mainly in the cerebellum (Figure 2B–D). The titer of Gabrd in the cerebellum of aged mice was about ten times higher than in the cortex, and more than 25 times higher than in the hippocampus (Figure 2B–D). Aging had no effect on Gabrd expression in the hippocampus and cerebellum, but in the cortex, the titer of the receptor was reduced more than three times by aging (Figure 2C).

## 7. GABA_b_–Gabbr

Gabbr is a heterodimeric metabotropic receptor formed by the Gabbr1 and Gabbr1 subunits, and its function is coupled to activation of a G protein and modulation of activities of downstream effectors such as adenylate cyclase [29]. We found that the expression of both subunits of the metabotropic receptor was decreased in the hippocampus of the aged animals (Figure 2B). In the cerebellum, the Gabbr amount was unaffected by aging. We could also observe a significant reduction in the sum of the Gabbr subunit concentrations in the cortex, although separately, the changes in the titers of the subunits were not statistically significant (Figure 2C).

## 8. Gad

Gad is an enzyme decarboxylating glutamate to produce GABA [30]. Our data show that the expression of the main isoform of Gad in the hippocampus, Gad2, was not affected by aging; however, the level of another Gad isoform, Gad1, was significantly elevated (Figure 2B). We could also observe significant increases in the titers of both Gad isoforms in the cerebellum of aged animals (Figure 2D), but there were no changes in Gad expression in the cortex of old mice (Figure 2C). 

## 9. Calcium/Calmodulin-Dependent Kinases

Activation of calcium/calmodulin-dependent protein kinase (Camk) is the first stage of transformation of calcium signaling into various forms of synaptic plasticity (for review, see [31,32]).

### 9.1. Camk1

The only isoform of Camk1 that we found in our analysis was Camk1d. We also observed some Camk1-associated peptides, but we were not able to annotate them precisely to selected Camk1 isoforms (they are described as “Camk1”, Figure 3A–C). In our analysis, we did not observe any significant aging-associated changes in the concentration of that group of Camk in the hippocampus and cortex (Figure 3A,B). On the other hand, the titer of Camk1 was significantly increased in the cerebellum of old mice (Figure 3C).

### 9.2. Camk2

Camk2 belongs to the most ubiquitous proteins in all brain structures [8,33]. We found that among Camk2 isoforms, the titer of Camk2a is the highest in all the studied brain structures (Figure 3A–D), and that its concentration in the hippocampus and cortex is many times higher than the titer of Camk2b, the second most abundant isoform of Camk2 (Figure 3A,B). Aging had no effect on the concentrations of the Camk2 isoforms, except for Camk2d, whose level was decreased in the hippocampus of old mice (Figure 3A).

### 9.3. Camk4

The molecular role of Camk4 in synaptic plasticity is different than Camk2. Active Camk4 localizes in the cell nucleus and regulates the transcription of genes involved in the late phase of memory formation [32].

Our study reveals that the concentration of Camk4 was significantly, almost two times, reduced in all the analyzed brain formations of old animals (Figure 3A–D). The protein titer was the lowest in the hippocampus and the highest in the cerebellum (Figure 3A–D). 

### 9.4. Camkk

Calcium/calmodulin-dependent protein kinase kinases (Camkk) phosphorylate and regulate the activity of Camk proteins. Our analysis revealed that the total concentration of Camkk was not statistically significantly affected by aging in the hippocampus and cortex (Figure 3A,B,D). In the cerebellum, the level of the Camkk1 isoform was more than 3-fold increased in old animals, whereas the Camkk2 concentration was more than two times lower in aged animals (Figure 3C). 

## 10. Prka–cAMP-Dependent Protein Kinase (PKA)

Prka is a conserved serine protein kinase with a wide distribution and relatively low specificity, which can phosphorylate various subunits of the AMPA and NMDA receptors and modulate their function [34]. 

### 10.1. PKA Catalytic Subunits—Prkac

Our analysis demonstrated that in the hippocampus, the most abundant catalytic subunits of PKA are Prkaca and Prkacb (Figure 4A). The concentrations of individual subunits were not affected by aging; however, the total Prkac protein concentration was slightly but statistically significantly decreased in aged mice (Figure 4A,D). Similar to the hippocampus, in the cortex and cerebellum, the most ubiquitous catalytic subunits of PKA were Prkaca and Prkacb; however, the concentrations of the proteins did not differ between young and old animals (Figure 4B,C).

### 10.2. PKA regulatory subunits—Prkar

We found that the overall expression of Prkar isoforms was the highest in the cortex, while in the hippocampus and cerebellum, the level of Prkar was similar (Figure 4A–D). Aging had no effect on the total concentration of Prkar, and the only age-related difference was a reduced level of Prkar2b in the hippocampus of old mice (Figure 4A).

## 11. Mitogen-Activated Protein Kinases—Mapk

Mapk proteins regulate a broad spectrum of cytoplasmic and nuclear processes involved in neuronal plasticity [35].

The results presented here demonstrate that Mapk1 was the most abundant Mapk in the hippocampus (Figure 4A). The total Mapk concentration in the hippocampus was not affected by aging (Figure 4A,D); however, Mapk3 expression was about 23% higher in aged animals (Figure 4A), and Mapk15 was increased almost seven times in aged hippocampi (Figure 4A). 

Both in the cortex and in the cerebellum, the total Mapk concentration was not affected by aging, and Mapk1 was the main kinase expressed in the two brain structures (Figure 4B,C).

## 12. Cholinergic Transmission

Acetylcholine transmission is mediated by two classes of receptors: the nicotinic receptors, which are acetylcholine-gated ion channels for sodium cations, and muscarinic receptors, which are metabotropic receptors coupled to the activity of trimeric G proteins.

Unexpectedly, in our study, we were not able to detect peptides which could be unequivocally attributed to nicotinic receptors. On the other hand, we determined the titer of several members of muscarinic cholinergic receptors.

### 12.1. Muscarinic Receptors—Chrm

In the hippocampus and cortex, but not in the cerebellum, we measured the titer of Chrm1, Chrm3 and Chrm4 (Figure 5A–C). In both brain formations, Chrm1 was the most abundant muscarinic receptor subunit. Aging reduced the level of Chrm1 and Chrm3 in the hippocampus, whereas in the cortex, the concentrations of Chrm3 and Chrm4 were decreased (Figure 5A,B).

### 12.2. Acetylcholine Metabolism

Choline acetyltransferase (Chat) is an enzyme involved in acetylcholine synthesis. We found that its expression was 2-fold increased in the hippocampus of aged animals (Figure 5A). In contrast to Chat, the titer of acetylcholine esterase (Ache), an enzyme degrading the neurotransmitter, was not affected by aging in this brain structure (Figure 5A). 

In the cortex, the concentrations of Chat and Ache were significantly higher than in the hippocampus and cerebellum; however, they were not influenced by aging (Figure 5A–C). 

We also did not observe age-related changes in the concentrations of Chat and Ache in the cerebellum, where the amount of the proteins was the lowest from all the analyzed brain regions. 

### 12.3. Vesicular Acetylcholine Transporter—Slc18a3

Slc18a3 is a protein responsible for acetylcholine loading into synaptic vesicles. Our study reveals that the concentration of Slc18a3 was the highest in the cortex of young mice (Appendix A), and that aging reduced the level of Slc18a3 by about 80% (Figure 5B). In the hippocampus, the protein amount was not affected by aging (Figure 5A), and in the cerebellum, the protein was detected only in aged animals (Appendix A). 

## 13. Monoamines Receptors, Signal Transmission and Metabolism

### 13.1. Receptors

Although we could measure the titer of several membrane proteins involved in monoamine signaling (e.g., in monoamines reuptake), we detected only a few members of monoamine receptors. We did not find peptides specifically attributed to dopamine receptors, and among numerous groups of serotonin receptors, we were able to unequivocally measure the titer of only one serotonin receptor (Htr1a) in the hippocampus of old animals (Appendix A).

In the hippocampus, we measured the titer of two members of alpha-adrenergic receptors: α2a (Adra2a) and α2c (Adra2c) (Figure 6A). Their concentration was lower in old animals; however, the changes were not statistically significant. In the cortex, Adra2c was present both in young and in old animals, but Adra2a was expressed only in young animals (Figure 6B). In the cerebellum, the only detected alpha-adrenergic receptor was Adra2a in young animals (Figure 6C). In this study, we were not able to unequivocally assign any peptides to beta-adrenergic receptor proteins, Adrb.

In contrast to beta-adrenergic receptors, we found a significant decrease in the expression of kinases involved in the desensitization of these receptors, Adrbk1, in the hippocampus and cerebellum (Figure 6A–C). This suggests that beta-adrenergic receptors may be ubiquitously expressed in the brain, but because of the methodology used in our experiment, the peptides related to Adrb receptors could not have been assigned to the proteins.

### 13.2. Monoamine Reuptake

The concentration of the serotonin transporter (Slc6a4) responsible for the neurotransmitter reuptake was present in all the studied brain structures and was practically unaffected by aging (Figure 6A–C). We observed a statistically significant increase in Slc6a4 in the hippocampus of aged mice; however, this increase was very low (Figure 6A). In contrast to the hippocampus, we found about a 3.5 times higher titer of Slc6a4 in aged cerebella, but the increase was not statistically significant (Figure 6C).

We determined the titer of a dopamine transporter (Slc6a3) in the cortex (Figure 6B). Its concentration was not affected by aging. 

## 14. Monoamine Deactivation

### 14.1. Monoamine Oxidase—Mao

Monoamine oxidase catalyzes the deamination of amines and is involved in the degradation of monoamines released by neurons and glia cells. There are two isoforms of Mao, Maoa and Maob, whose expression is attributed, respectively, to neurons and glial cells [36].

The results of our study reveal that both Mao isoenzymes were ubiquitously expressed in all the brain formations (Figure 6A–C). In the hippocampus, the concentration of Maoa was reduced by aging, while the level of Maob was significantly elevated in the aged animals (Figure 6A). The same trend could be observed for the Mao isoforms in the cortex (Figure 6B). 

In the cerebellum, the titer of Maoa was unaffected by aging, but the concentration of Maob was more than four times higher in old animals than in young ones (Figure 6C).

### 14.2. Catechol O-Methyltransferase—Comt

Catechol O-methyltransferase catalyzes O-methylation and, thus, the inactivation of monoamines such as adrenaline, dopamine and serotonin. In our analysis, we found that the enzyme was relatively abundant in all the studied brain formations, and that its level was not affected by aging (Figure 6A–C).

## 15. Monoamine Synthesis

### 15.1. Tryptophan Hydroxylase 2—Tph2

Tryptophan hydroxylase is the enzyme catalyzing the first step of serotonin synthesis. Our study shows that the level of Tph2 in all the brain structures of old animals was similar (Figure 6A–C). In the hippocampus of young animals, the titer of Tph2 was much lower than in other brain formations, but aging resulted in a significant, more than 8-fold, increase in the titer of the enzyme in this structure (Figure 6A). In the cortex and cerebellum, aging had no effect on the expression of the enzyme (Figure 6B,C). 

### 15.2. Tyrosine Hydroxylase—Th

Th is involved in the first step of monoamine (such as dopamine, adrenaline and noradrenaline) synthesis from tyrosine. The titer of Th was the highest in the cortex (Figure 6C), and the presence of the enzyme was not detected in the hippocampus (Figure 6A). Aging had no effect on the expression of Th in both brain formations (Figure 6B,C). 

### 15.3. Aromatic-L-amino-acid decarboxylase—Ddc

Ddc (also known as DOPA decarboxylase and AADC, and 5-hydroxytryptophan decarboxylase) participates in neurotransmitter synthesis, catalyzing the decarboxylation of various substrates such as DOPA, phenylalanine, histidine and 5-hydroxytryptamine. In our studies, we found that Ddc was more than two times higher in the hippocampi of aged animals (Figure 6A), whereas in the cortex and cerebellum, the protein amount was not significantly modified by aging (Figure 6B,C). 

## 16. Signal Transduction

Stimulation of several metabotropic receptors is associated with modification of the activity of adenylate cyclase and/or phospholipases and changes in the concentration of secondary messengers such as cAMP and inositol trisphosphates.

### 16.1. Adenylyl Cyclase—Adcy

Adcy is the enzyme that catalyzes the formation of cAMP from ATP, and its activity is regulated after stimulation of metabotropic cholinergic and catecholaminergic receptors. Our study reveals that in the hippocampus, Adcy2 and Adcy9 were the most abundant isoforms of the cyclase (Figure 6A). We found that the concentration of the main form of the enzyme in the hippocampus, Adcy9, was significantly increased in old mice, while the titer of Adcy1, Adcy3, Adcy6 and Adcy8 decreased in old animals (Figure 6A). Aging had no effect on the abundance of the main forms of Adcy, Adcy5 and Adcy9, in the cortex, but it reduced the titer of Adcy1, Adcy2, Adcy3 and Adcy6 (Figure 6B). We also observed a reduction in the main Adcy isoform in the cerebellum, Adcy (Figure 6C).

### 16.2. Phospholipase C—Plc 

Phospholipase C (Plc), an enzyme that hydrolyzes phospholipids, is involved in intracellular signal transmission after stimulation of the muscarinic cholinergic receptor (Chrm) and alpha-adrenergic receptor (Adra1) [37].

In our study, we found several members of the Plc class in all the studied brain structures (Figure 6A–C). The most abundant Plc isoforms in the hippocampus were Plcg1 and Plch2, whose titer was not affected by aging (Figure 6A). In contrast to the hippocampus, almost all Plc isoforms, except for Plcd3, were downregulated in the cortex of aged mice (Figure 6B). In the cerebellum, Plcb4 was the predominant isoform of Plc, but its titer was unaffected by aging (Figure 3C). The only Plc whose concentration was different in the cerebellum of old mice was Plch2 (Figure 6C).

## 17. Cytomatrix Active Zone—CAZ

The CAZ is a presynaptic region involved in neurotransmitter release [38]. Proteins within this region mediate, directly and indirectly, synaptic vesicle fusion with the presynaptic membrane. Among this group of proteins, several functional classes may be distinguished: SNAREs (soluble N-ethylmaleimide-sensitive factor attachment protein receptors), which can be divided into v-SNAREs (vesicle-associated SNAREs) and t-SNAREs (target membrane-associated SNAREs), and proteins involved in calcium-mediated docking of synaptic vesicles [39].

In our analysis, we did not observe numerous significant age-related changes in the concentration of CAZ proteins in the hippocampus and cerebellum (Figure 7A,C). The only age-affected proteins in the hippocampus were Snap25 and Vap, whose titers were lower in old mice (Figure 7A). On the other hand, the concentration of several CAZ proteins was significantly modified, usually elevated, by aging in the cortex (Figure 7B). The most prominent changes were related to v- and t-SNARE proteins such as syntaxins (Stx), synaptotagmine (Syt), synaptobrevins (Vamp), synaptogyrins (Syngr) and synaptophysins (Syp) (Figure 7B). This was accompanied by an increase in the concentrations of proteins involved in the calcium-dependent machinery of synaptic vesicle docking and anchoring, such as syntaxin-binding proteins (Stxbp) and synucleins (Snc) (Figure 7B). Detailed information on titers of the CAZ proteins is provided in Appendix A.

## 18. Postsynaptic Density—PSD

The postsynaptic density is the protein-rich region attached to the postsynaptic membrane where proteins involved in signal reception and transmission as well as modulation of synaptic plasticity are located [40]. In our study, we found several members of PSD proteins in all the studied brain structures (Appendix A), and changes in the titers of some of them (receptors, proteins involved in the synthesis of secondary messengers, etc.) are described in previous sections of the paper.

The concentration of PSD proteins in the cerebellum was not significantly altered by aging. Psd and Shank3 were the only exceptions: their titer was, respectively, reduced and elevated in old animals (Figure 7C). In contrast to the cerebellum, aging significantly reduced the concentration of several members of PSD proteins in the cortex and hippocampus (Figure 7A,B). Among them were such proteins important for synaptic plasticity as Dlg4/Psd95, Syngap1, Shank1 and Shank2 in the cortex (Figure 7B), and Psd, Dlg, Dlgap and Shank in the hippocampus (Figure 7A). A detailed list of these changes is provided in Appendix A.

## 19. Trans-Synaptic Cell Adhesion Molecules—CAMs

Trans-synaptic cell adhesion molecules regulate synaptic plasticity via organization of the synaptic connection, control synapse morphology and regulate receptor functions [41].

We quantitatively measured the titers of almost 140 proteins annotated to the trans-synaptic cell adhesion molecules (Appendix A). Overall, the level of most of the CAMs was affected by aging in the hippocampus and cortex (Figure 8A,B), but relatively small changes were observed in the cerebellum (Figure 8C). We observed a significant decrease in the concentration of several cell adhesion molecules such as cadherins (Cdh), catenins (Ctnn), ephrins (Eph), receptor-type tyrosine-protein phosphatase (Ptpr), neurexins (Nrxn) and the Lin7 protein, both in the hippocampus and in the cortex of old animals (Figure 8A,B). The full set of proteins presented in Figure 8 is described in Table 1, and the titer of various isoforms of Cdh, Ctnn, Eph and Nrxn is presented in Appendix A.

The most significant changes in the cerebellum were associated with the expression of Lgi, whose concentration was more than two times higher in old mice (Figure 8C). Lgis are secreted proteins regulating receptor distribution and cellular interactions in the nervous system [73]. Although in the cerebellum, the titer of most of the cell adhesion proteins was unaffected by aging, the titer of some of them such as ephrins (Eph), liprin-alpha (Ppfia) and neurexins (Nrxn) was reduced (Figure 8C). We also found a significant elevation in the Lgi protein in the hippocampus and cortex (Figure 8A–C). The roles of individual CAMs are summarized in Table 1.

## 20. Extracellular Matrix (ECM) Perineuronal Net (PNN) Proteins

PNNs are extracellular matrix structures which cover the surface of the neuronal cell body and protrusions in the central nervous system and stabilize synapses in adult brains. PNNs are composed mainly of chondroitin sulphate proteoglycans [74]. 

We found more than 70 members of the PNNs in young and old murine brain structures (Appendix A). Generally, we observed elevated levels of PNN proteins in the aged brains, and these increases were attributed mainly to an enormous increase in Hapln (hyaluronan and proteoglycan link protein), the main component of PNNs (Figure 9A–C). Except for Hapln, the titer of collagens (Col) was increased in the hippocampus and cerebellum, and laminins (Lam) were elevated in all the studied brain structures (Figure 9A–C). Unexpectedly, we found that some of the PNN proteins were downregulated in aged mice, e.g., the concentrations of talin-2 (Tln2) and semaphorins (Sema) in the hippocampus and cortex (Figure 9A,B).

On analyzing the ECM protein, we found an interesting dependence: the titers of most large proteoglycans (such as Agrn, Ncan, Vcan and Bcan) were significantly increased in the hippocampus and cerebellum of old animals, but not in the cortex (Figure 9A–C). The cortex also appears to be the most stable brain formation in the context of the expression of ECM remodeling enzymes. We found several members of ADAM proteases in all the studied brain structures. Only the ADAM23 concentration was elevated by aging in the cortex (Figure 9B), ADAM10, ADAM11 ADAM22 and ADAM23 were increased in the hippocampus of old mice, while the titers of ADAM22 and ADAM23 were elevated in the cortex (Figure 9A,C).

Among metalloproteinases, we could measure the concentration of only Mmp17, and it was slightly reduced by aging in the hippocampus and cerebellum (Figure 9A–C). The roles of CAM protein groups are summarized in Table 2.

## 21. Discussion

Aging-associated changes in the protein composition of brain formations are still far from being well understood. Several valuable pieces of data have been delivered by studies using immunocytochemical and immunohistochemical techniques. They demonstrated a diverse expression of several proteins in various brain structures, and even in different populations of neurons [97,98,99]. However, because of the methodological limitations, such studies have been restricted to a small number of proteins, and they could deliver only semiquantitative data on protein expression. They also could not deliver the real concentration of proteins, given in absolute values (e.g., in mol/g protein), in the studied samples.

In this paper, we used a mass spectrometry-based technique, the label-free total protein approach method, to quantitatively describe proteins involved in signal transmission in the hippocampus, cerebral cortex and cerebellum of young and old mice. All the brain formations are heterogeneous structures composed of various cells, among which neurons and astrocytes are the most abundant. Due to that, the results presented in this paper cannot be unequivocally annotated to neurons or glial cells. However, in several cases, the expression of the analyzed proteins is known to be related almost exclusively to one type of cell, e.g., GABA-synthetizing enzymes in the hippocampus, which are associated mainly with interneurons (for review, see [100]).

Our analysis demonstrated that the molecular machinery involved in the excitatory and inhibitory transmission (respectively, glutamatergic and GABAergic) in the hippocampus and cortex was significantly altered (reduced) by aging. In turn, the expression of glutamate and GABA receptors in the cerebellum was practically unchanged; however, the titer of GADs (the enzymes involved in GABA synthesis) was strongly elevated in aged mice. 

The decreased concentration of proteins involved in glutamatergic and GABAergic transmission might indicate a decreased neuronal plasticity of the hippocampus and cortex in aged animals. 

The decreased titer of Camk4 might be the next marker of the lower synaptic plasticity of old animals as Camk4 is a protein that is indispensable for the formation of long-term memory [101]. We observed not only a significant reduction in Cam4 in all the studied brain formations but also a very high level of this kinase in the cerebellum, which is in line with studies showing the substantial role of active Camk4 for cerebellar long-term depression, regarded to be the main form of synaptic plasticity in this brain structure [102,103,104]. We did not find any age-related differences in the titer of Camk2, which is known to be directly involved in synaptic enhancement. However, Camk2 is a protein expressed at a very high level and involved in a variety of cellular events, and hence the lack of statistically significant changes in the whole structures is not unexpected.

We did not observe numerous changes in the concentration of other kinases involved in synaptic transmission and plasticity, such as PKA and Mapk. However, we found that the concentration of Prkac, a catalytic subunit of PKA, is significantly reduced by aging in the hippocampus. This might suggest lower excitatory transmission and plasticity of aged hippocampi because PKA-dependent phosphorylation of the AMPA subunits directly controls the synaptic incorporation of AMPA receptors [105].

Unexpectedly, we were not able to measure the titers of nicotinic acetylcholine receptors (as well as dopamine and serotonin receptors, except for Htr1a). Since we identified several other membrane proteins, the lack of these receptors in our analysis is not an effect of the sample preparation method but results from an actual absence of unique peptides that could be unequivocally annotated to those receptors.

In contrast to nicotinic receptors, we identified muscarinic acetylcholine receptors (Chrm) in the hippocampus and cortex, and we found that their titer was significantly reduced in old animals. Interestingly, the concentration of the enzyme Chat involved in acetylcholine synthesis was significantly increased in the hippocampus and cerebellum, and the cerebellum was the only brain structure in which we were not able to measure the Chrm concentration. 

The most pronounced difference between young and old animals regarding adrenergic transmission was the very high increase in the titer of monoamine oxidase b, an enzyme responsible for the deactivation of amines. This may suggest that adrenergic transmission and, overall, the catecholaminergic signaling are reduced in old animals. However, we also found a significant aging-associated reduction in the Adrbk1 concentration, a kinase that is involved in the desensitization of adrenergic receptors. The downregulation of Adrbk1 should lead to an increase in the sensitivity of adrenergic receptors—an adaptation to reduced amounts of neurotransmitters caused by a strongly increased activity of Moab. 

The capacity to release neurotransmitters also depends on the presence of proteins involved in synaptic vesicle trafficking. Our study demonstrates that the concentration of proteins engaged in neurotransmitter release, such as v-SNAREs, t-SNAREs and exocytosis-associated proteins, was relatively constant during aging in the hippocampus and cerebellum. In contrast, the titer of proteins involved in neurotransmitter release was significantly elevated in old cortices, which may suggest that the amount of active synapses in old cortices is higher than that in young cortices. However, the increase in the presynaptic part of the neurotransmitter release apparatus in the cortex was not correlated with an elevation in the postsynaptic proteins forming the machinery of signal reception and transmission. We observed a significant reduction in the titer of these proteins both in the cortex and in the hippocampus.

As synaptic plasticity also depends on the expression of proteins organizing synapse morphology, we checked the concentration of the trans-synaptic cell adhesion molecules and extracellular matrix proteins that form perineuronal nets.

We found that the titers of most CAMs were decreased in aged hippocampi and cortices, while in the cerebellum, the concentrations of these proteins were relatively stable. Although the expression of several members of CAMs has been studied in the context of Alzheimer’s disease [106,107,108], our analysis provides the first global picture of age-dependent changes in CAM expression. In contrast to almost all other CAMs, we found a significant increase in the Lgi protein level. Lgi proteins are known to participate in synapse formation and maturation but also in the myelination process (for review, see [42]). Crucial Lgi binding partners in synapse development are ADAM proteins such as ADAM11, ADAM22 and ADAM23 [42]. In our studies, we detected significant increases in the concentrations of all these ADAM isoforms in the hippocampus. This finding suggests a higher number of mature, stable synapses in aged hippocampi than in young hippocampi.

In contrast, we found that the majority of the perineuronal net proteins were more abundant in the aged brain formations. The differences were most pronounced for the Hapln, Acan and Bgn proteins, whose titers were elevated in all the brain structures. 

Moreover, we also observed a significant elevation in proteoglycans in the hippocampus and cerebellum. This is in apparent opposition to previous immunohistochemical studies which suggested that in the mouse brain, chondroitin sulfate proteoglycans that mainly comprise the Hapln protein do not change with aging [109]. This contradiction may result from the different age of the young animals used in this (1 month old) and the previous study (4 months old) [109]. Such an interpretation is roughly in line with observations that the expression of some proteoglycans steadily increases in the rat brain up to 5 months of age, but then the titer of some proteins decreases (for review, see [110]).

Although we found that some proteins or protein group concentrations were affected by aging in a similar manner in all the studied brain formations, “simultaneous aging” does not seem to be a rule. We observed that the cerebellar proteome displayed the smallest number of changes during aging, while the hippocampal and cortical proteomes were unstable.

Concluding, our analysis is the first in-depth and comprehensive quantitative proteomic study describing changes in the concentration of proteins critical for signal transmission and synaptic plasticity in the hippocampus, cortex and cerebellum of young and old mice. The data presented here provide a general picture of the effect of physiological aging on synaptic plasticity and might suggest potential drug targets for anti-aging therapies.

## 22. Conclusions

Aging is known to change brain functions, and our studies demonstrate that these changes are related to an altered expression of several receptors, signal transduction proteins and structural proteins involved in synapse formation. Aging-related changes in the proteome were observed in all the examined brain structures, e.g., in the hippocampus, cerebral cortex and cerebellum. With aging, the most stable brain structure is the cerebellum, while the hippocampus and cortex exhibit a similar amount of differentially expressed proteins involved in neurotransmission and neuroplasticity. Our study reveals that there is no single universal pattern of aging-related changes in proteomes; instead, each of the analyzed brain formations represents its own mode of changes.

## Figures and Tables

**Figure 1 cells-10-02021-f001:**
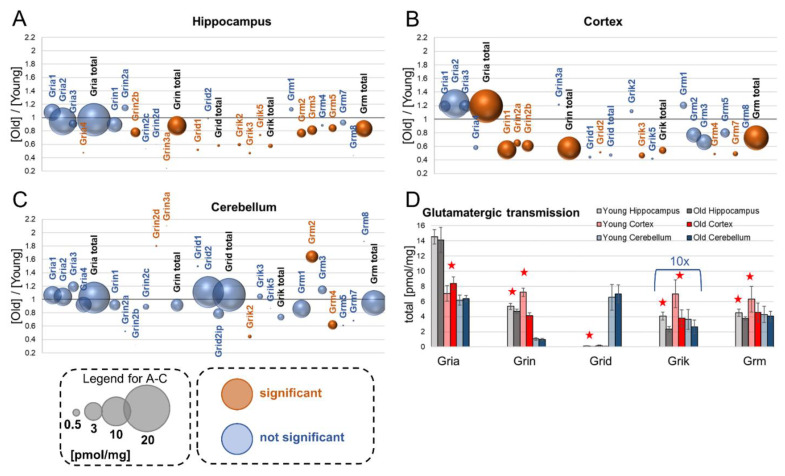
Changes in proteins involved in glutamatergic transmission upon aging. Plots show ratios of protein concentrations in old vs. young brain structures: hippocampus (**A**), cortex (**B**) and cerebellum (**C**). Total concentrations of various receptor families in young and old animals are shown in (**D**) (★ *p* < 0.05). The size of bubbles is proportional to the average protein concentration in the respective old brain structures. Statistically significant differences are shown as the red bubbles.

**Figure 2 cells-10-02021-f002:**
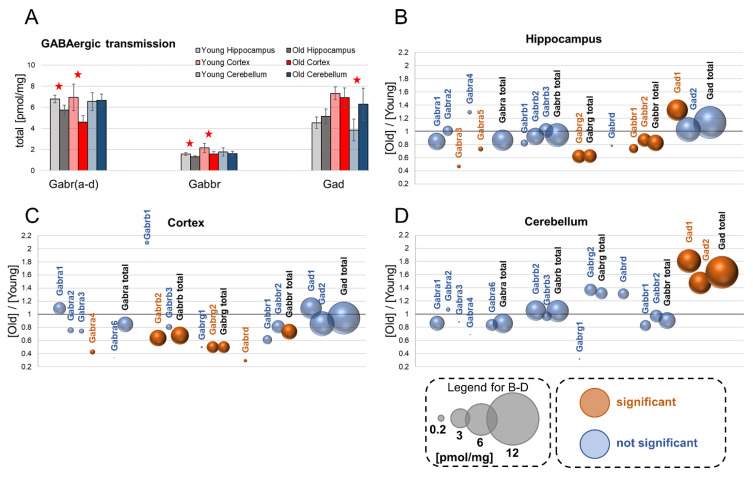
Changes in proteins involved in GABAergic transmission upon aging. Total concentrations of various receptor families in brains structures of young and old animals (**A**) (★ *p* < 0.05). Plots (**B–D**) show ratios of protein concentrations in old vs. young brain structures: hippocampus (**B**), cortex (**C**) and cerebellum (**D**). The size of bubbles is proportional to the average protein concentration in the respective old brain structures. Statistically significant differences are shown as the red bubbles.

**Figure 3 cells-10-02021-f003:**
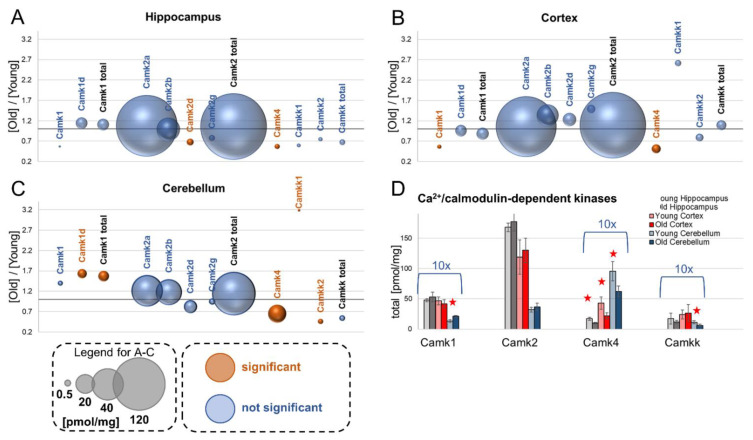
The effect of aging on calcium/calmodulin-dependent kinases. The ratios of protein titers in old vs. young brain structures in: hippocampus (**A**), cortex (**B**) and cerebellum (**C**). Total concentrations of the kinase families in young and old animals are shown in (**D**) (★ *p* < 0.05). The size of bubbles is proportional to the average protein concentration in the respective old brain structures. Statistically significant differences are shown as the red bubbles.

**Figure 4 cells-10-02021-f004:**
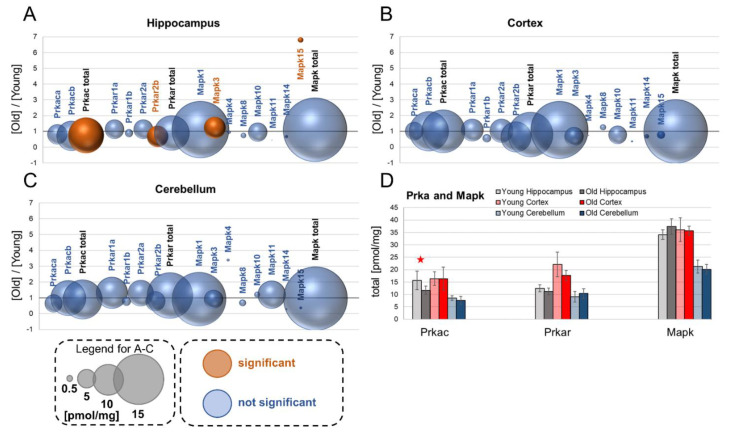
Aging-related changes in PKA (Prka) and Map kinases. The ratios of the kinase concentrations in old vs. young brain formations in: hippocampus (**A**), cortex (**B**) and cerebellum (**C**). Total concentrations of the kinase families in young and old animals are shown in (**D**) (★ *p* < 0.05). The size of bubbles is proportional to the average protein concentration in the respective old brain structures. Statistically significant differences are shown as the red bubbles.

**Figure 5 cells-10-02021-f005:**
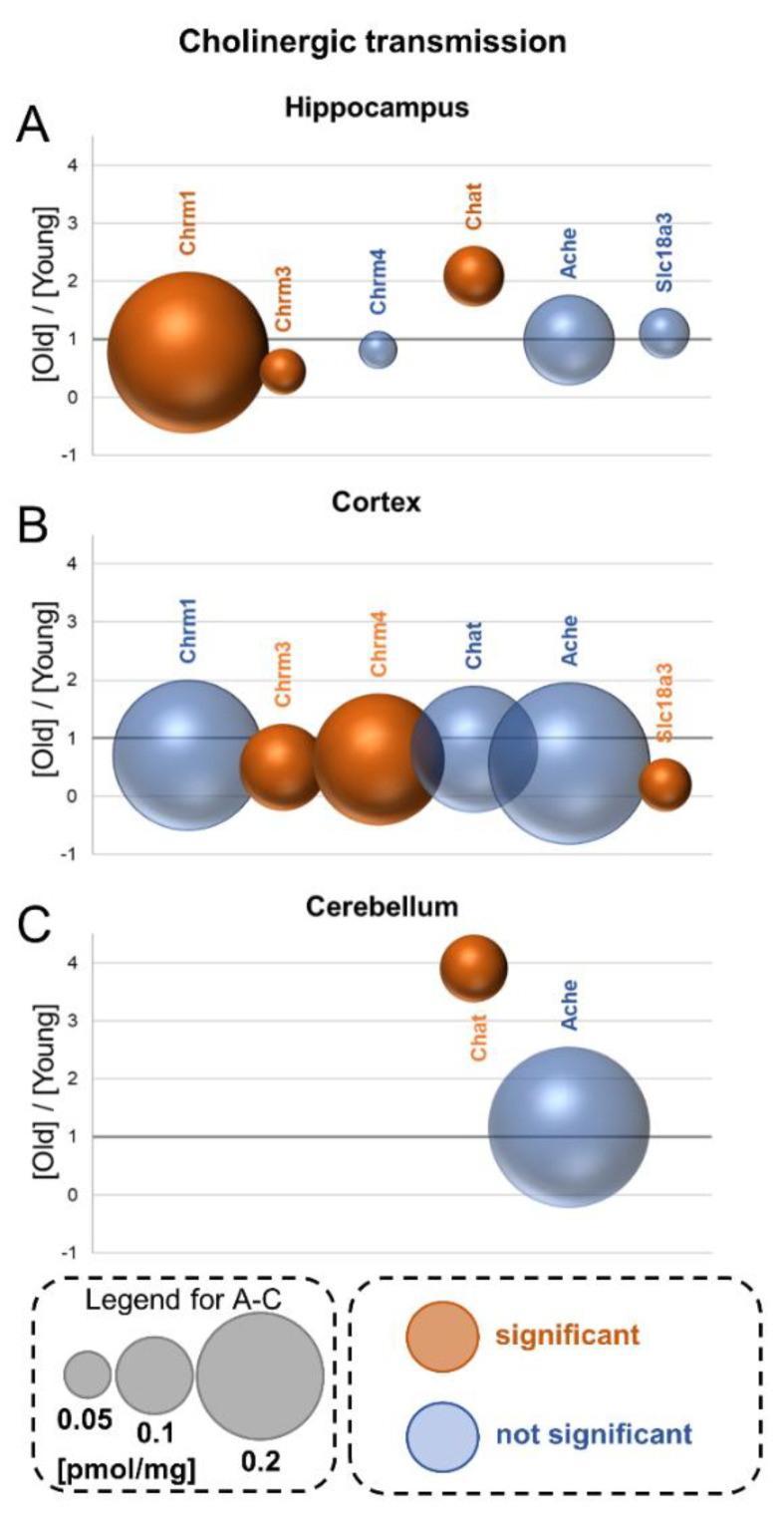
Aging-associated changes in cholinergic transmission. Plot shows the ratios of the titers of the receptors and proteins involved in acetylcholine metabolism in old vs. young brain formations in: hippocampus (**A**), cortex (**B**) and cerebellum (**C**). The size of bubbles is proportional to the average protein concentration in the respective old brain structures. Statistically significant differences are shown as the red bubbles.

**Figure 6 cells-10-02021-f006:**
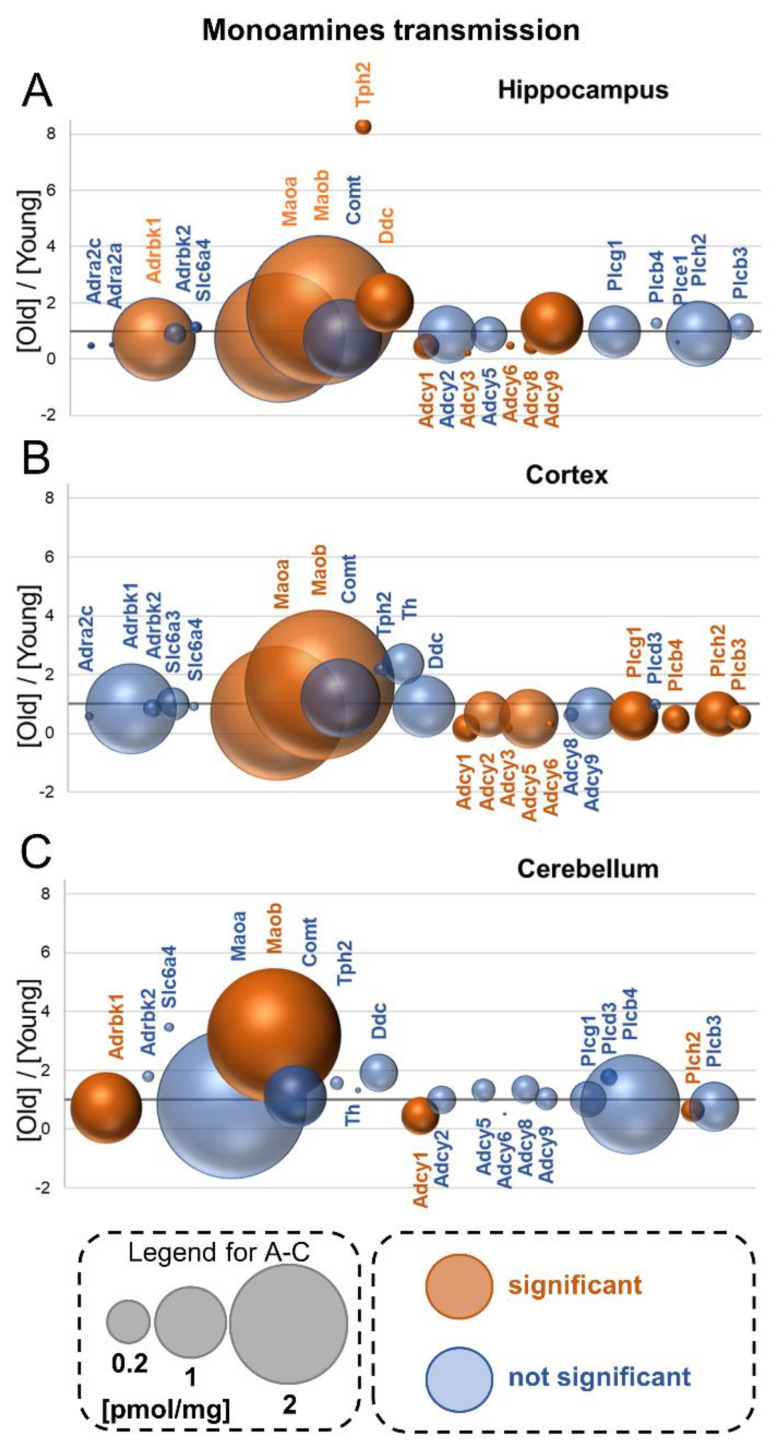
Aging-associated changes in monoamine transmission and metabolism, and in intracellular machinery of signal transduction. Plot shows the ratios of the proteins in old vs. young brain formations in: hippocampus (**A**), cortex (**B**) and cerebellum (**C**). The size of bubbles is proportional to the average protein concentration in the respective old brain structures. Statistically significant differences are shown as the red bubbles.

**Figure 7 cells-10-02021-f007:**
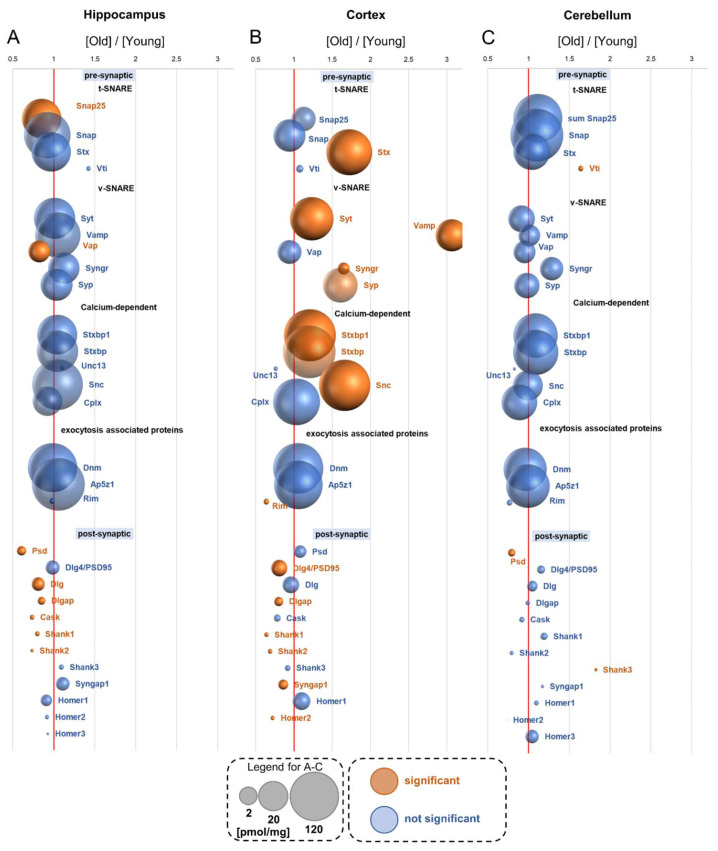
The effect of aging on proteins of the cytomatrix active zone and postsynaptic density. Plot shows the ratios of the proteins in old vs. young brain formations in: hippocampus (**A**), cortex (**B**) and cerebellum (**C**). The size of bubbles is proportional to the average protein concentration in the respective old brain structures. Statistically significant differences are shown as the red bubbles.

**Figure 8 cells-10-02021-f008:**
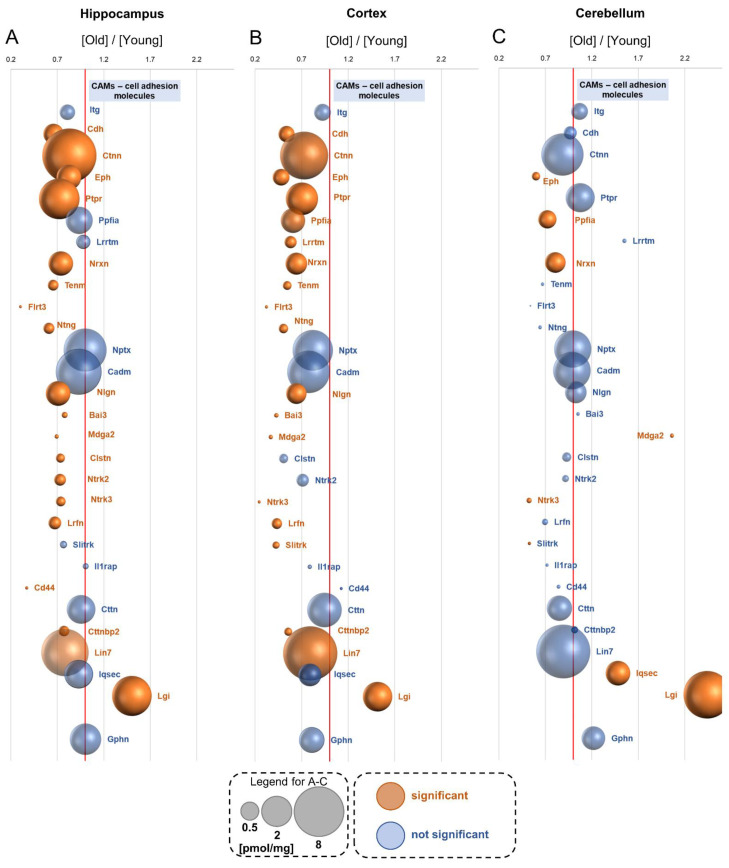
Changes in proteins of the trans-synaptic cell adhesion molecule group upon aging. Plot shows the ratios of the proteins in old vs. young brain formations in: hippocampus (**A**), cortex (**B**) and cerebellum (**C**). The size of bubbles is proportional to the average protein concentration in the respective old brain structures. Statistically significant differences are shown as the red bubbles.

**Figure 9 cells-10-02021-f009:**
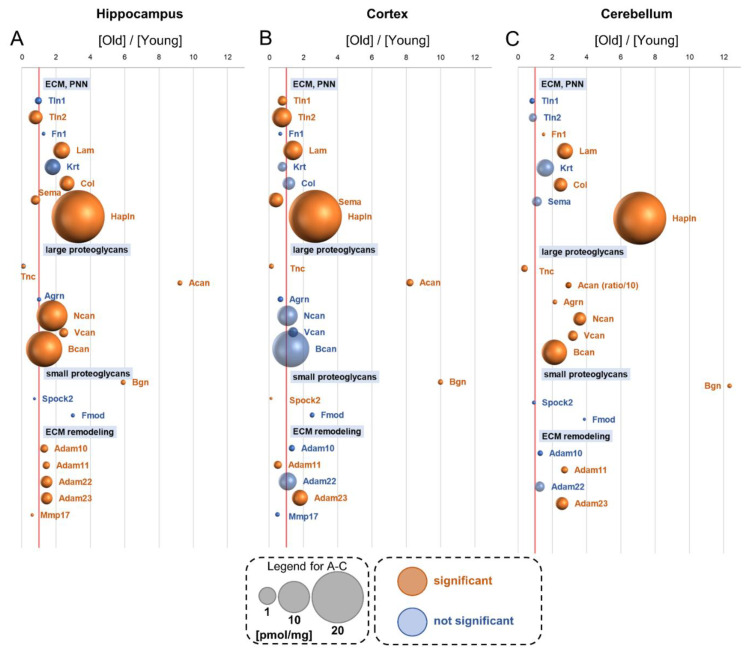
Changes in proteins of the extracellular matrix and perineuronal net group upon aging. Plot shows the ratios of the proteins in old vs. young brain formations in: hippocampus (**A**), cortex (**B**) and cerebellum (**C**). The size of bubbles is proportional to the average protein concentration in the respective old brain structures. Statistically significant differences are shown as the red bubbles.

**Table 1 cells-10-02021-t001:** CAMs, or cell adhesion molecules, involved in synapse formation, restructuring and stabilization.

Gene Names	Protein/Protein Group	Synaptic Localization	CAM Partners	Reference
*Itg*	Integrins	Pre and Post	AMPA receptors, Laminin, Talin, Vinculin, Shank	[42]
*Cdh*	Cadherins	Pre and Post	AMPA receptors	[43]
*Ctnn*	Catenins	Pre and Post	PSD, Cadherins, Catenins	[44]
*Eph*	Ephrins	Pre and Post	Fibronectin, Rho GTPases, NMDA receptors, Ephr	[45]
*Ptpr*	Receptor-type tyrosine-protein phosphatases	Post	Glutamate receptors, TrkC, SALMs, Netrin-G	[46]
*Ppfia*	Liprin-alpha	Pre	SNARE complex	[47]
*Lrrtm*	Leucine-rich repeat transmembrane neuronal proteins	Post	Neurexin, HSPGs (heparan sulphate proteoglycan)	[48,49]
*Nrxn*	Neurexins	Pre and Post	Neuroligins, Dystroglycan, Lrrtms, GABA_A_ receptors, Latrophilins	[50]
*Tenm*	Teneurin	Pre	Latrophilins (LPHNs), Dystroglycans	[51]
*Flrt3*	Leucine-rich repeat transmembrane protein FLRT3	Pre	Adgrl, Unc5B	[52]
*Ntng*	Netrin	Pre	AMPA receptors, Netrin-G ligand	[53]
*Nptx*	Neuronal pentraxin	Pre	AMPA receptors	[54]
*Cadm*	Cell adhesion molecules (SynCAMs)	Pre and Post	Other Cadm	[55]
*Nlgn*	Neuroligin	Post	Neurexin	[56]
*Bai3*	Brain-specific angiogenesis inhibitor 3	Post	Neurexin, Glutamate receptors	[57]
*Mdga2*	MAM domain-containing glycosylphosphatidylinositol anchor protein 2	Post	Neuroligin	[58]
*Clstn*	Calsyntenins	Post	Membrane trafficking proteins	[59]
*Ntrk2*	BDNF/NT-3 growth factor receptor	Post	BDNF, PSD, Ntfk2	[60]
*Ntrk3*	NT-3 growth factor receptor	Post	TrkA	[61]
*Lrfn*	Leucine-rich repeat and fibronectin type-III domain (SALMs)	Pre and Post	Ptpr	[62]
*Slitrk*	SLIT and NTRK-like proteins	Post	Ptpr	[63]
*Il1rap*	Interleukin-1 receptor accessory protein	Post	Ptpr	[64]
*Cd44*	CD44 antigen	Post	Hyaluronan, Collagen, Growth factors, Cytokines, ADAM 17, Rho GTPases	[65]
*Cttn*	Src substrate cortactins	Post	Clathrin	[66]
*Cttnbp2*	Cortactin-binding protein 2	Post	Contractin, Potassium channels	[67]
*Lin7*	Protein lin-7	Post	BDNF, Potassium channels	[68]
*Iqsec*	IQ motif and SEC7	Post	PSD, AMPA receptors, Gephyrin	[69]
*Lgi4*	Leucine-rich repeat LGI Lgi4	Pre and Post	Mielin	[70]
*Lgi1*	Leucine-rich repeat LGI Lgi1	Pre and Post	Potassium channels, ADAM22, PSD95	[71]
*Gphn*	Gephyrin	Post	GABA_A_ receptors	[72]

**Table 2 cells-10-02021-t002:** ECM, or extracellular matrix, proteins involved in cell adhesion and, thus, regulation of neuronal development, axon guidance, synapse formation (maturation) and neuronal plasticity.

Gene Names	Protein/Protein Group	Activity	References
*Tln*	Talin	Adhesion	[75]
*Fn1*	Fibronectin	Adhesion	[76]
*Lam*	Laminins	Adhesion	[77]
*Krt*	Keratins	Adhesion	[78]
*Col*	Collagens	Adhesion	[79]
*Sema*	Semaphorins	Adhesion	[80]
*Hapln*	Hyaluronan and proteoglycan link proteins	Adhesion	[81]
*Tnc*	Tenascin	Neurite outgrowth	[82]
*Acan*	Aggrecan core protein	Adhesion	[83]
*Agrn*	Agrin	Adhesion	[84]
*Ncan*	Neurocan core protein	Adhesion	[85]
*Vcan*	Versican core protein	Adhesion	[86]
*Bcan*	Brevican core protein	Adhesion	[87]
*Bgn*	Biglycan	Adhesion	[88]
*Spock2*	Testican-2	Adhesion	[89]
*Fmod*	Fibromodulin	Adhesion	[90]
*Adam10*	Disintegrin and metalloproteinase domain-containing protein 10	Digestion	[91]
*Adam11*	Disintegrin and metalloproteinase domain-containing protein 11	Neural adhesion and axon guidance	[92,93]
*Adam22*	Disintegrin and metalloproteinase domain-containing protein 22	Neural adhesion	[93,94]
*Adam23*	Disintegrin and metalloproteinase domain-containing protein 23	Neural adhesion neurite outgrowth	[93,95]
*MMP17*	Matrix metalloproteinase-17	Digestion	[96]

## Data Availability

The data presented in this study are available on request from the corresponding author.

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
