# Peer review of "Quantitative Proteomics Reveals Significant Differences between Mouse Brain Formations in Expression of Proteins Involved in Neuronal Plasticity during Aging"

_cells, 2021, doi:10.3390/cells10082021_

Round 1

Reviewer 1 Report

The manuscript by Drulis-Fajdasz et al have quantitatively evaluated the levels of brain proteins in 1 and 24 months old mice. The levels of different proteins were measured using the label-free proteomics approach. The findings of the study suggest that protein homeostasis is differentially unperturbed in the hippocampus and cortex but not in the cerebellum. My major comments for the manuscript is given below:

Major Comments:

  1. A recent study has evaluated structural changes in mouse brains using in vivo MRI. Although mouse brain volume was found to be almost stable at three weeks, the thickness of the cerebral cortex kept decreasing with maximal changes during the first three months. Moreover, myelination is still increasing between three and six months, however, most dramatic changes are over by three months. These results emphasize that mice should be at least three months old when adult animals are needed for brain studies. Hence, the use of one-month-old mice in the current study seems to be inappropriate to understand brain protein changes with aging. The changes in brain protein level from 1 to 24 months may be confounded by development and aging.    
  2. The methods used in the study are too brief for replication. It needs to be expanded.
  3. The expression used for the quantification of protein level is not clear. The meaning of various terms in the expression for protein quantification is not clear, and required further clarification.
  4. The result section is too big, and needs to be concise.
  5. There is no information regarding the reproducibility of repeated measurements for the quantified protein level, i.e. levels of different proteins estimated in a given sample.
  6. The finding of the proteomics study should be validated by immunohistochemical analysis of some proteins.
  7. A thorough review of the manuscript for the formation of a sentence, etc. is required. e.g. The statement on Line # 65-68 “Using the label-free Total Protein Approach (TPA) method animals and we measured the titer of more than 7,000 proteins in each of the studied brain region [13] and in this paper, we present so far the most in-depth quantitative proteomic description synaptic plasticity-related changes during the physiological aging of mice.” is too long and complicated and need to be broken in several small sentences. Line#85 “The lysates were obtained as described in [13].

Reviewer 2 Report

In the submitted manuscript the Mass Spectrometry–based technique was used to asses a proteome analysis on different brain area during aging. The manuscript is overall well done and data are robust. I only have the following few queries

Slightly revise English text (as for example, 65-67 “Using the label-free Total Protein Approach (TPA) method animals and we measured the titer of more than 7,000 proteins in each of the studied brain region [13] and in this 66 paper, we present so far the most in-depth quantitative proteomic description synaptic 67 plasticity-related changes during the physiological aging of mice”. there is something missing in this semntence).

Please describe how the different brain areas were dissected, including a picture with cut sites. In particular, describe how hippocampus was dissected and what part was used for the study (as a whole or only partially?)

Due to brain sex dimorphism, why female mice only were used in the study?

Why sexually immature mice were used as young controls?

How many targets were further validated by Western blot?
